# Blood-Epigenetic Biomarker Associations with Tumor Immunophenotype in Patients with Urothelial Carcinoma from JAVELIN Bladder 100

**DOI:** 10.3390/cancers17142332

**Published:** 2025-07-14

**Authors:** Thomas Powles, Srikala S. Sridhar, Joaquim Bellmunt, Cora N. Sternberg, Petros Grivas, Ewan Hunter, Matthew Salter, Ryan Powell, Ann Dring, Jayne Green, Alexandre Akoulitchev, Roy Ronen, Janusz Dutkowski, Robert Amezquita, Chao-Hui Huang, Diane Fernandez, Robbin Nameki, Keith A. Ching, Jie Pu, Michelle Saul, Shibing Deng, Alessandra di Pietro, Craig B. Davis

**Affiliations:** 1Barts Cancer Institute, Experimental Cancer Medicine Centre, Queen Mary University of London, St. Bartholomew’s Hospital, Charterhouse Square, London EC1M 6BQ, UK; 2Princess Margaret Cancer Centre, University of Toronto, Toronto, ON M5G 2M9, Canada; 3Department of Medical Oncology, Dana-Farber Cancer Institute, Harvard Medical School, Boston, MA 02215, USA; 4Englander Institute for Precision Medicine, Division of Hematology and Medical Oncology, Meyer Cancer Center, Weill Cornell Medicine, New York, NY 10021, USA; 5Fred Hutchinson Cancer Center, University of Washington, Seattle, WA 98109, USA; 6Oxford BioDynamics Plc, Oxford OX4 2WB, UK; 7Data4Cure, Inc., Waltham, MA 02451, USA; 8Pfizer, San Diego, CA 92121, USA; 9Pfizer, Tucson, AZ 85714, USA; 10Pfizer srl, 20152 Milano, Italy

**Keywords:** urothelial carcinoma, immune checkpoint inhibitors, avelumab, chromatin conformation, epigenetics, spatial transcriptomics

## Abstract

The JAVELIN Bladder 100 trial investigated avelumab maintenance treatment in people with advanced bladder cancer. In this exploratory analysis of data from the JAVELIN Bladder 100 trial, researchers investigated the association between chromatin conformation markers in peripheral blood specimens and tumor gene expression. Researchers identified a distinct set of chromatin conformation markers that were associated with high vs. low immune gene expression. The loci covered by these markers contained genes participating in multiple pathways that may have direct and indirect effects on immune responses. These findings support the use of blood chromatin conformation assays to query biological states in tissues.

## 1. Introduction

Exploratory analyses of biomarkers from clinical trials of immune checkpoint inhibitors (ICIs) have highlighted diverse biological pathways associated with response to ICI treatment; however, identifying biomarker assays that provide optimal predictive response value remains a major challenge [1,2]. In the phase 3 JAVELIN Bladder 100 trial, avelumab first-line (1L) maintenance plus best supportive care (BSC) significantly prolonged overall survival (OS) and progression-free survival (PFS) vs. BSC alone in patients with advanced urothelial carcinoma (UC) without disease progression after 1L platinum-based chemotherapy [3]. Based on these results, avelumab 1L maintenance was established as a standard of care in patients with advanced UC [4,5].

Exploratory tumor biomarker analyses from JAVELIN Bladder 100 revealed complex relationships with treatment outcomes [2]. Assays for individual biomarkers used in clinical practice, such as programmed death ligand 1 (PD-L1) and tumor mutation burden (TMB), did not optimally identify patients who were significantly more likely to benefit from avelumab 1L maintenance. Further analyses of gene mutation and expression within tumor biopsies identified 3 classes of biological pathways that may interact with avelumab treatment effect: (1) increased tumor immunogenicity, including APOBEC3-associated mutagenesis; (2) increased activity of both innate and adaptive immune pathways; and (3) decreased expression of genes associated with signaling pathways linked to tissue growth, such as Notch, angiogenesis, and TGF-β. These results suggested that optimization of the avelumab treatment effect would require deeper analysis of biomarkers capturing the functional heterogeneity within the tumor microenvironment.

As highlighted in the recently updated cancer immunity cycle [6], non-transformed cell types in the tumor microenvironment, including stromal and myeloid cells, have emerged as important modulators of anti-tumor immunity released by ICIs. These findings, made possible through the application of novel technologies, such as spatial transcriptomics, support hypotheses that response to ICI may be contingent on host factors such as germline single nucleotide polymorphisms [7], microbiomes [8], and epigenetic modification of the 3-dimensional (3D) structure of chromatin [9,10,11,12,13]. EpiSwitch is an established 3D genomic biomarker platform, based on the original chromosome conformation capture approach, that has demonstrated promising utility in various disease settings [14,15,16,17,18,19], including response to avelumab observed in a retrospective study [20].

We applied the EpiSwitch platform to peripheral blood of patients enrolled in the JAVELIN Bladder 100 trial to explore the potential interplay between epigenetic host factors and immune activity in the tumor microenvironment. A 3-stage process was used: (1) identification of chromatin conformation markers (CCMs) associated with tumor immune gene expression signature; (2) statistical modeling to rank CCMs association with TMB and OS following avelumab plus BSC vs. BSC alone; and (3) characterization of the expression and function of genes covered by the highest-ranked CCMs.

## 2. Materials and Methods

### 2.1. Description of Specimens Collected for Analysis

JAVELIN Bladder 100 (NCT02603432) was a randomized phase 3 trial evaluating avelumab 1L maintenance (administered every 2 weeks) plus BSC vs. BSC alone in patients with advanced UC whose cancer had not progressed after 4–6 cycles of 1 L platinum-based chemotherapy [3]. Patients received treatment until disease progression, unacceptable toxicity, or withdrawal of consent. OS (primary endpoint) and PFS (secondary endpoint) were significantly prolonged with avelumab plus BSC vs. BSC alone. Peripheral blood was collected from patients 4–10 weeks after the date of administration of the last dose of chemotherapy but before treatment with avelumab plus BSC or BSC alone. Blood samples from 496 patients satisfied quality and consent criteria and were subjected to further evaluation, as shown in Appendix A. Characteristics and outcomes in the biomarker analysis set compared with the full analysis set are shown in Appendix A, respectively. The biomarker analysis set did not differ notably from the full analysis set.

### 2.2. Identification of Candidate CCMs Using the EpiSwitch Platform

#### 2.2.1. Preparation of 3D Genomic Templates

EpiSwitch 3D libraries, with chromosome conformation analytes converted to sequence-based tags, were prepared from frozen whole-blood samples using EpiSwitch protocols following the manufacturer’s instructions for EpiSwitch Explorer Array kits (Oxford BioDynamics Plc, Oxford, UK). Samples were processed on the Freedom EVO 200 robotic platform (Tecan Group Ltd., Männedorf, Switzerland). Briefly, the whole-blood sample was diluted and fixed with a formaldehyde-containing EpiSwitch buffer. Density cushion centrifugation was used to purify intact nuclei. Following a short detergent-based step to permeabilize the nuclei, restriction enzyme digestion and proximity ligation were used to generate the 3D libraries. Samples were centrifuged to pellet the intact nuclei before purification with an adapted protocol from the QIAamp DNA FFPE Tissue kit (Qiagen, Hilden, Germany), eluting in 1× TE buffer (pH 7.5). Three-dimensional libraries were quantified using the Quant-iT PicoGreen dsDNA Assay kit (Invitrogen, Waltham, MA, USA) and normalized to 5 ng/mL of input DNA prior to interrogation by PCR with Eppendorf Mastercycler Pro S (Eppendorf, Hamburg, Germany).

#### 2.2.2. Array Design

Custom microarrays were designed using the EpiSwitch pattern recognition algorithm, which operates on Bayesian modeling and provides a probabilistic score for a region’s involvement in long-range chromatin interactions. The algorithm was used to annotate the GRCh38 human genome assembly across approximately 1.1 million sites with the potential to form long-range chromosome conformations. The most probable interactions were identified and filtered on probabilistic score and proximity to protein, long non-coding RNA, or microRNA coding sequences. Predicted interactions were limited to EpiSwitch sites that were ≥10 kb and <300 kb apart. Repeat masking and sequence analysis were used to ensure unique CCM sequences for each interaction. The EpiSwitch Explorer Array (Oxford BioDynamics Plc, Oxford, UK, Product Code X-HS-AC-02) containing 60-mer oligonucleotide probes was designed to interrogate potential 3D genomic interactions. In total, 964,631 experimental probes and 2500 control probes were added to a 1 × 1 comparative genomic hybridization microarray slide design. The experimental probes were placed on the design in singlicate, with the controls in groups of 250. The control probes consisted of 6 different EpiSwitch interactions that are generated during the extraction processes and used for monitoring library quality. An additional 4 external inline control probe designs were added to detect a non-human (*Arabidopsis thaliana*) spike in DNA added during the sample labeling protocol to provide a standard curve and control for labeling. The external spike DNA consists of 400 bp single-stranded DNA fragments from genomic regions of *Arabidopsis thaliana*. Array-based comparisons were performed as described previously, with the modification of only 1 sample being hybridized to each array slide in the Cy3 channel.

#### 2.2.3. Translation of Array-Based 3D Genomic Markers to PCR Readouts

Stepwise diagnostic biomarker discovery process using EpiSwitch technology and data analysis: The EpiSwitch technology platform (Oxford Biodynamics, Oxford, UK) pairs high-resolution chromosome conformation capture results with regression analysis and a machine learning algorithm to develop disease classifications [21,22,23]. To select epigenetic biomarkers, patient samples for each class were screened for statistically significant differences in conditional and stable profiles of genome architecture. All samples used for the nested PCR biomarker discovery were processed per the manufacturer’s instructions using the proprietary EpiSwitch reagents and standard protocols [24]. A PCR product was detected only if the corresponding CCM was detected. To account for variable marker stability in assay conditions, 3 template concentrations (1×, 2×, and 4×) were evaluated.

*Genomic mapping.* Mapping was performed using the Bedtools closest function [21] for the 3 closest protein-coding loci: upstream, downstream, and within the long-range chromosome interaction (Gencode v33). All markers were visualized using the EpiSwitch analytical portal.

#### 2.2.4. Identification of Top 25 CCMs Associated with Tumor JAV-Immuno Scores

Analyses for marker prioritization were performed using libraries developed for the R Statistical Language (R version 4.1.2). JAV-Immuno includes the following genes: *CCL5*, *CD2*, *CD244*, *CD247*, *CD3E*, *CD3G*, *CD6*, *CD8B*, *CD96*, *CST7*, *EOMES*, *GFI1*, *GPR18*, *GRAP2*, *IL7R*, *ITK*, *KCNA3*, *KLRD1*, *NLRC3*, *PRF1*, *PSTPIP1*, *SH2D1A*, *SIT1*, *THEMIS*, *TRAT1*, and *XCL2*. Candidate markers were identified using EpiSwitch whole-genome arrays comparing 20 patients whose tumors were JAV-Immuno^hi^ (greater than the median) and 20 patients whose tumors were JAV-Immuno^lo^ (less than the median). The data were normalized, quality controlled, and statistically analyzed using the EpiSwitch array pipeline powered by the LIMMA package. The top 150 EpiSwitch markers were selected based on their corrected *p*-value (false discovery rate) and directionality/interaction frequency for JAV-Immuno. Feature engineering with the selected EpiSwitch CCMs was performed using the same samples used in the array discovery, with an additional 20 samples each of the JAV-Immuno^hi^ and JAV-Immuno^lo^ classes. A reduced set of 25 CCMs was generated using the univariate EXACT test (contingency testing), binary weighting, multivariate analyses, permutated glmnet, VSURF, Boruta, and varSelRF. An XGBoost algorithm model was used for further refinement of the CCM list [22]. Shapley additive explanations values, used to characterize the contribution of each feature in a model to the final prediction and final model selection, were used to confirm marker prioritization [23].

### 2.3. Prioritization of Candidate CCMs Based on Estimated Interactions with TMB and OS

The starting feature set comprised 25 CCMs, each measured at 3 dilution levels for a total of 75 features, and was assessed in 496 patients for its association with OS.

#### Model Development

In step 1, features were filtered out prior to modeling if <40 subjects were included in either the present or absent categories and <15 subjects in any of the 4 subgroups (avelumab present, avelumab absent, BSC present, BSC absent). This filter removed 15 features.

In step 2, due to high correlation among CCMs measured at multiple dilution levels, further screening was performed to either select 1 dilution level per CCM or remove the CCM from subsequent multivariate modeling. The screening assessed each CCM in association with OS using 3 nested Cox proportional hazards models. The first model assessed the CCM main effect (direct association with OS); the second model included the CCM 2-way interaction with treatment and OS as well as the main effect; the third model incorporated the 3-way interaction among CCM, treatment, and TMB in addition to the main effect and 2-way interaction. For each CCM, this resulted in 3 models for each dilution level. The *p*-values derived from the 3-way interaction, 2-way interaction, and main effect terms were sequentially evaluated for each CCM. The dilution level with the lowest 3-way interaction *p*-value of <0.1 was selected. If none were <0.1, the process was repeated for the 2-way interaction and main effect *p*-values. If none of these *p*-values were <0.1, then the CCM was eliminated from downstream modeling.

Finally, in step 3, the remaining 16 features were fed into an elastic net Cox proportional hazards model-building procedure to select a final feature set. The elastic net [25], a regularized regression method using a weighted sum of the L1 and L2 penalty terms for the lasso [26] and ridge [27] regression, was used for statistical model building (R package version 4.1.2: glmnet). The hyperparameters α and λ, which control the weight applied to each of the penalty terms and the amount of shrinkage on the coefficients, were tuned using repeated 5-fold cross-validation (CV). For each round of 5-fold CV, a pair of α and λ associated with the optimal model performance (C-index) was determined (Appendix A). The same procedure was iterated 1000 times using different random data partitions to account for potential CV error. The most frequently selected hyperparameter pairs were chosen as optimal and applied to the full dataset to obtain the final models. All but 1 of the 16 input features were retained by this process, generating the list of 15 CCMs (Appendix A).

### 2.4. Assessment of Potential CCM Gene Expression

Public data on genes labeled using the Human Genome Organization nomenclature have been compiled into a knowledge graph [28] by Data4Cure (www.data4cure.com; accessed on 30 March 2025). Knowledge graph associations between genes and specific biomedical entity domains (e.g., cell types or pathways) summarize evidence from thousands of publicly available datasets and millions of publications linking biomedical entities.

### 2.5. Visium Data Processing

SpaceRanger 2.0.1 was used to align raw reads to the GrCh38 genome and feature count generation using the Human Transcriptome probe set v2.0. Visium data resolution was enhanced 6-fold by inferring pseudospots using the BayesSpace method [29]. Briefly, using the BayesSpace R package Visium Space Ranger, the outs folder was processed with seed 102, spatialPreprocess(sceobj, platform = “Visium”, n.PCs = 15, n.HVGs = 2000, log.normalize = T), followed by spatialCluster(sceobj, q = 8, d = 15, platform = “Visium”, init.method = “mclust”, model = “t”, gamma = 1, nrep = 10,000, burn.in = 100, save.chain = F), and then spatialEnhance(sceobj, q = 8, platform = “Visium”, d = 15, model = “t”, gamma = 1, jitter_prior = 0.3, jitter_scale = 3.5, nrep = 100,000, burn.in = 100, save.chain = F). Enhanced features and spatial coordinates for the full expression matrix were computed using enhanceFeatures and output for subsequent visualization using the Data4Cure platform [30] (www.data4cure.com; accessed on 30 March 2025).

For the expression signatures of each pseudospot, the Data4Cure platform was used to compute the gene signature score as the mean normalized average expression value across the signature genes. The spot-level signature scores were plotted heat map style using the pseudospot coordinates and overlaid on the approximate corresponding region of the H&E image as a background image. An unmarked, blinded H&E image was annotated by a pathologist for lymphoid aggregates (numbered and circled in green).

## 3. Results

### 3.1. Selection of a Tumor Immune Gene Expression Signature to Screen for Candidate CCMs

The JAVELIN Bladder 100 study included 496 patients with tumor transcriptome data whose blood could be analyzed by EpiSwitch (Appendix A). A 26-gene signature (JAV-Immuno), identified in JAVELIN Renal 101 and validated in JAVELIN Bladder 100 [2,31], displayed modulation of the interaction between TMB, OS, and treatment (avelumab plus BSC vs. BSC alone) comparable to that of PD-L1 (Figure 1A). In the avelumab plus BSC arm, a significant OS benefit was associated with JAV-Immuno levels (high vs. low) in patients with low TMB (HR, 0.52 [95% CI, 0.351–0.773]; *p* = 0.0012) but not in patients with high TMB (HR, 1.22 [95% CI, 0.756–1.957]; *p* = 0.42). No association was found between either biomarker and OS in the BSC-alone arm. JAV-Immuno was therefore chosen as the measure of tumor immune response to probe for CCMs.

### 3.2. Screening and Selection of Candidate CCMs Through Assessment of Potential Interactions with Treatment Effect and TMB as Well as Tumor Phenotype

The CCM discovery process was conducted as previously described (Figure 1B; see Methods for details) [32]. Patient samples were classified as JAV-Immuno^hi^ if the corresponding tumor expression was greater than the sample median and JAV-Immuno^lo^ if it was less than or equal to the median. Nuclei templates from 20 JAV-Immuno^hi^ and 20 JAV-Immuno^lo^ specimens were hybridized to an array of >900 K oligonucleotides spanning the human genome, thereby identifying 150 loci that could be either positively or negatively associated with the JAV-Immuno classification. Nested polymerase chain reaction (PCR) assays designed to give a positive reading only if the PCR primer targets were juxtaposed by chromatin conformation were applied to 40 JAV-Immuno^hi^ and 40 JAV-Immuno^lo^ samples (Appendix A). These nested PCR data were used as a training set to build JAV-Immuno^hi/lo^ classifier models. PCR data from the remaining 416 samples were used to refine and test the classifier models. The models with optimal test accuracy (0.68) contained 25 CCMs (Figure 1C). Univariate analysis of CCM association with JAV-Immuno scores indicated that individual CCMs could be either positively or negatively associated with JAV-Immuno, depending on the detection titer dilution, suggesting complex relationships between the CCMs and expression of individual genes. The candidate CCMs were further evaluated for association with TMB and OS using Cox proportional hazards and elastic net models (see Section 2—Materials and Methods for details). The 15 CCMs selected in the final model were subjected to further biological annotation (Appendix A).

### 3.3. Potential Cell Types and Pathways Represented by the Genes Covered by the CCMs

CCMs in the final model were distributed across the human genome (Appendix A). These CCMs were detected in <20% to >70% of the patients in the analysis population (Appendix A), and no linkage disequilibrium was observed (Appendix A). Twelve of the 15 CCMs covered distinct regions, and 3 overlapped in a ≈290 kb region on chromosome 22:20707691–20999032 (Appendix A). Forty-four major genes in the Human Protein Atlas [33] were mapped to these 15 CCMs, of which 29 genes (66%) were covered by >1 CCM in the final model. In contrast, only 95 of the 323 genes (29%) mapping to the remaining CCMs in the original set were covered by >1 CCM. Thus, the final model was enriched for loci that were covered by multiple CCMs, consistent with possible functional relevance. Only one of the genes in the original 26-gene JAV-Immuno signature (*CD6*) [31] was present in the 367 genes mapped to the original set, and it was not present in the 15 CCMs from the final model. Individual genes from these 15 CCMs (Table 1) showed both positive and negative associations with JAV-Immuno (Figure 1D), suggesting that host factors may either enhance or diminish anti-tumor immunity.

We evaluated potential expression of genes mapping to the 15 CCMs using the Tabula Sapiens single-cell transcriptomic atlas [34], which contains single-cell sequencing data from almost 500,000 cells, representing over 400 cell types and 4 major cell lineages (immune, endothelial, stromal, and epithelial) assessed in approximately 22 distinct anatomical locations collected from multiple donors. Appendix A shows expression patterns by cell type for 6 genes mapping to chromosome 22:20707691-20999032, which was covered by 3 CCMs in the final model (Appendix A). Substantial variability in gene expression across cell types was observed, suggesting that cell-specific gene-regulatory machinery operates in conjunction with regulation of chromatin conformation.

We performed a comprehensive analysis of associations between genes and cell types reported in the literature and public databases using the Data4Cure knowledge graph and platform [30]. Genes in the JAV-Immuno signature were predominantly associated with lymphocytes, including T cells, B cells, and natural killer (NK) cells (Figure 2A), as previously described [31]. Genes covered by the selected CCMs were associated with endothelial and fibroblast cell types, in addition to lymphocytes (Figure 2B). Myeloid lineages, including monocytes, neutrophils, and macrophages, were associated with genes in both signatures (Figure 2A,B). JAV-Immuno genes were predominantly associated with Gene Ontology Biological Processes related to lymphocyte activation (Figure 2C) [35]; in contrast, genes mapping to the selected CCM set were associated with a wide range of biological processes (Figure 2D). These findings suggest that the chromatin conformations represented by the selected CCMs may exert both distal and proximal effects on lymphocyte activation.

### 3.4. Use of Spatial Profiling to Locate CCM Genes Within the TME

Spatial transcriptomic data were used to explore the topographic expression of genes represented by JAV-Immuno and the selected CCMs. A procured bladder cancer specimen was selected for initial evaluation due to its robust representation of immune and non-immune cell types [2]. Figure 3(A-i) shows a hematoxylin and eosin (H&E)–stained image of the specimen identifying the position of lymphoid aggregates. Spatial indexing of transcriptomic data from this specimen identified 8 clusters by BayesSpace (Appendix A). Similarities in the histology and distribution of predefined gene signatures (Appendix A) suggested that the 8 clusters represented 4 functional regions, labeled as epithelial, stromal, transitional, and lymphoid aggregates (Figure 3(A-ii)). Figure 3(B-i) depicts the spatial distribution of genes characteristic of tertiary lymphoid structures (TLS), originally identified in patients with melanoma treated with ICIs [36]. The highest overall expression of the component genes was observed in the functional region labeled as lymphoid aggregates (Figure 3(B-ii,B-iii)), suggesting that this region could be a locus of immune activation and maturation in the tumor microenvironment. Enrichment of gene expression signatures characteristic of T follicular helper cells, mature B cells, cytotoxic T-cells, and dendritic cell recruitment further support the lymphoid aggregate regions as a site for priming and maturation of anti-tumor lymphocytes (Appendix A).

The spatial distribution of genes comprising the JAV-Immuno signature was strongly correlated with those in the TLS signature, suggesting that JAV-Immuno is distinctly associated with TLS function (Figure 3(C-i–C-iii), Appendix A). To assess the potential generalizability of these findings, expression of JAV-Immuno genes was evaluated in a collection of specimens representing TLS from bladder, renal, and breast cancers. All 25 genes in the JAV-Immuno signature were positively associated with the presence of TLS in these regions (Figure 3(C-iv)). Overall, JAV-Immuno appears to be more closely associated with TLS than other signatures associated with ICI response [2,37,38,39,40].

The expression of 33 of the 35 major genes mapped to the 15 CCMs was assessed in the bladder specimen and appeared to be distributed broadly throughout the specimen, with slightly lower overall expression in the stromal region (Figure 3(D-i)). Hierarchical clustering of the genes by distribution between the 4 annotated regions identified subsets of genes that were prominent in the lymphoid aggregate and epithelial regions, as well as the stromal region (Figure 3(D-ii,D-iii)). Ten of the 15 selected CCMs in Appendix A contained ≥1 gene that was more highly expressed in the lymphoid aggregate region relative to the epithelial region (Figure 3(D-ii)).

As for JAV-Immuno (Figure 3B), potential generalizability of the CCM findings was assessed by evaluating expression of the genes in the TLS specimen collection (Figure 3(D-iv)). Several genes positively associated with JAV-Immuno (Figure 1D) and expressed in the lymphoid aggregate region (Figure 3(D-ii)) were associated with TLS regions in the collection (Figure 3(D-iv)), including *POU2F2*, *MBNL1*, *ABI3BP*, *VPS13C*, and *CPEB1*. Other genes, including *CMPK2, RSAD2*, *RNF144A*, and *SUCNR1*, were positively associated with JAV-Immuno but were expressed more prominently outside the lymphoid aggregate regions (Figure 3(D-ii)) and were not as strongly associated with TLS in the collection (Figure 3(D-iv)). Participation in antiviral and antibacterial responses has been described for *RSAD2* [41,42] and *CMPK2* [42,43]. Reduced levels of *RNF144A* have been associated with PD-L1 stabilization and tumorigenesis in mouse bladder cancer models [44] and poor prognosis in patients with breast cancer [45]. *SUCNR1* has been linked to altered immune infiltration of tumors in ovarian cancer [46] and renal cancer [47]. The adaptive immune processes represented by JAV-Immuno may thus be tuned by responses to stimuli in the tumor microenvironment that are outside of the lymphoid aggregates.

### 3.5. Specific Assessment of Associations Between a CCM-Covered Gene, TMB, and Avelumab Treatment Effect

Potential relationships between genes covered by the CCMs with TMB and OS with avelumab plus BSC were assessed using a 3-way interaction test (Figure 4A). This analysis was conducted in genes that were expressed in >50% of tumor samples and/or had a coefficient of variation of >5% (Appendix A). The strongest positive monogenic association was seen with *POU2F2*, which also showed a positive association with JAV-Immuno (Figure 1D). *POU2F2* is a transcription factor involved in B-cell differentiation and collaboration between B-cells and T-cells and is expressed in multiple cell types involved in antigen presentation, including B-cells and monocytes (Figure 2B) [48,49]. The presence of the corresponding CCM was associated with low *POU2F2* gene expression and decreased expression of *POU2F2* target genes, suggesting that the CCM negatively regulates *POU2F2* and downstream processes, including generation of mature B cells and TLS formation (Figure 4B).

We then explored relationships between the *POU2F2* CCM, TMB, and OS in the avelumab plus BSC and BSC arms. *POU2F2* CCM interactions with TMB and OS were more pronounced in the avelumab plus BSC arm than in the BSC arm (Table 2). The absence vs. presence of the *POU2F2* CCM was associated with a positive effect on OS in patients treated with avelumab plus BSC whose tumors displayed low TMB (less than the median value of 7.66 nonsynonymous single nucleotide variants per megabase); median OS was 37 vs. 18 months (HR, 0.46 [95% CI, 0.24–0.89]; *p* = 0.02). These findings suggest that the effect of CCM modulation on *POU2F2* expression may be strongest when the treatment effect is contingent on immune responses to limited tumor immunogenicity.

## 4. Discussion

Large pivotal studies such as the phase 3 JAVELIN Bladder 100 trial are particularly valuable for investigating complex interactions between tumor and patient biomarkers in advanced UC, since (1) avelumab 1L maintenance was associated with significant OS and PFS benefit and (2) avelumab was compared with BSC rather than an active treatment control. Therefore, we utilized this study to explore the possible connections between tumor phenotype and peripheral blood epigenetic biomarkers detected by a clinically validated platform. These findings are subject to known caveats related to exploratory post hoc analyses, including the possibility of false discovery and the lack of statistical power to detect all meaningful associations; as such, they should be considered hypothesis-generating. Necessary next steps include (a) validation in independent datasets, (b) further investigation of potential clinical confounders such as age, sex, ECOG performance status, or tumor staging, (c) confirmation of generalizability and robustness given potential biases in the training set, and (d) prospective demonstration of utility in clinical studies before consideration for use in treating patients. Even so, this study lays a foundation for further identification and validation of CCM biomarkers for use in clinical development.

The associations between tumor phenotype, blood chromatin structure, and avelumab treatment effect are consistent with the hypothesis that systemic host chromatin structures may shape the tumor microenvironment and thereby influence the treatment effect of avelumab and other ICI therapies. Although the initial CCM screen evaluated associations with a gene signature of immune activity, including TLS formation, the emerging CCMs covered genes that are expressed by tumor cells (e.g., *RSAD2*, *CMPK2*) and stromal fibroblasts (e.g., *SUCNR1*) as well as immune cells (e.g., *POU2F2*). The observation that peripheral blood CCMs could be associated with genes in distinct regions of the TME has intriguing parallels to the observation, from tumor gene mutation and expression, that multiple pathways reflecting immune-tumor interactions were associated with avelumab treatment effect [2]. The generalizability of the distribution of these genes across immune, stromal, and tumor compartments needs to be confirmed via spatial profiling of additional representative tumor sets. Such investigations will clarify the potential contribution of systemic and tumor-specific biomarkers to ICI treatment optimization in UC, whether alone or in combination with other treatments.

The findings from these analyses open multiple avenues for further investigation and validation. The relationship between TMB, TLS formation in tumors, and POU2F2 gene expression may help to explain why the performance of TMB as an ICI treatment biomarker has been variable [2]. Similar investigations into specific contributions to anti-tumor immune function may be conducted for MBNL1, a pre-mRNA alternative splicing regulator [52,53]; ABI3BP, an extracellular matrix protein associated with tumor suppression and immune infiltration [54,55]; and VPS13C, a lipid transport protein participating in mitochondrial stability [56] and in associations between lysosomal function and innate immune activation [57]. Direct contributions of CMPK2 and RSAD2 to anti-tumor immune response induction, as opposed to serving as indirect biomarkers of interferon signaling, should be clarified. Potential contributions of other genes with currently poorly defined functions, such as ZNF family members and C2CD4A/B, remain to be fully assessed. Such investigations may further enable the appropriate interpretation and clinical validation of these biomarkers. Interactions between peripheral blood CCMs and tumor mutations or mutation burden that are also measurable in blood may provide novel opportunities for non-invasive monitoring of patient responses to ICI treatment.

## 5. Conclusions

A clinically validated CCM platform was applied to peripheral blood from patients with matched tumor data from a pivotal phase 3 study of ICI therapy for advanced UC. The emerging associations between host factors in peripheral blood and tumor molecular phenotype illustrate the utility of exploratory analyses from pivotal studies for biomarker-guided advances in treatment strategies for patients with advanced UC.

## Figures and Tables

**Figure 1 cancers-17-02332-f001:**
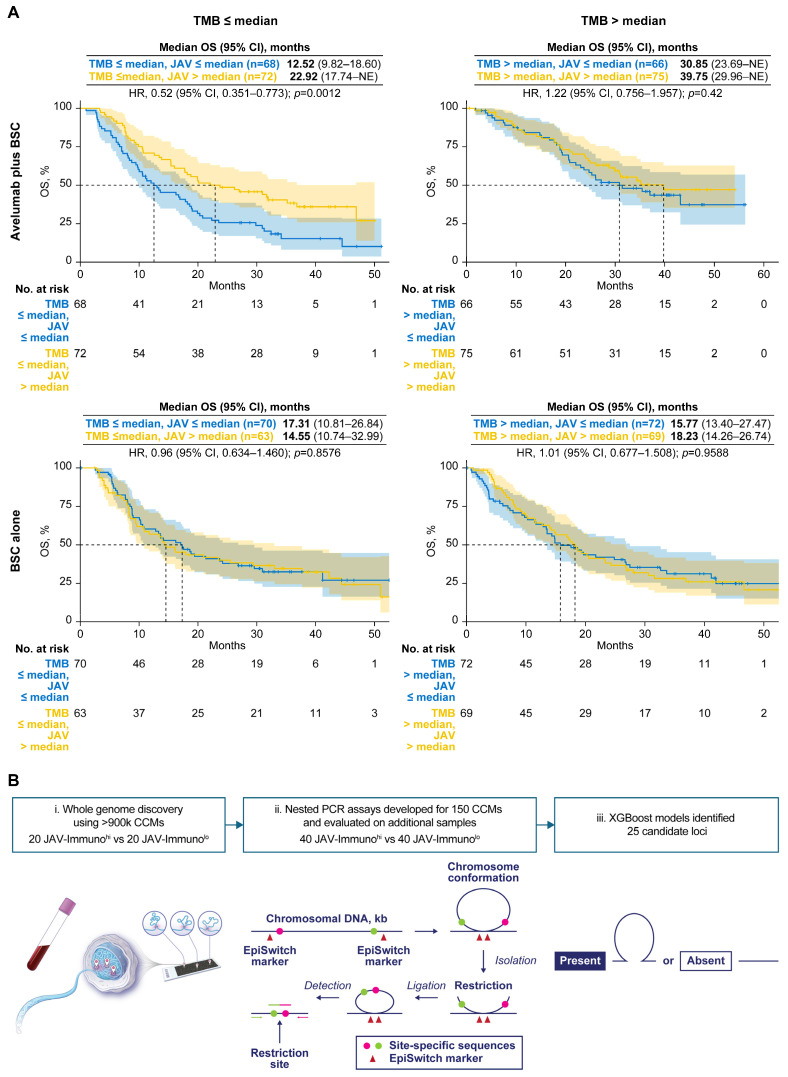
Blood chromatin markers associated with JAV-Immuno in tumor and interactions between TMB, treatment, and OS. (**A**). Modulation of TMB interactions with treatment based on OS outcome by JAV-Immuno score. Kaplan-Meier plots of high baseline TMB (≥median) vs. low baseline TMB (<median) are shown for subgroups defined by treatment (avelumab plus BSC or BSC) and baseline JAV-Immuno score^hi/lo^ (based on median split). A Cox proportional hazards model with no adjustment for baseline covariates was used. A 2-sided Wald test was used to determine *p*-values. (**B**). Discovery of CCMs using EpiSwitch platform. i. DNA from fixed and permeabilized nuclei was digested with restriction enzymes and hybridized to an oligonucleotide array designed to detect sequences brought in proximity by chromatin conformation. ii. Nested PCR assays were designed for 150 sequences that could potentially differentiate between patients with JAV-Immuno^hi^ and JAV-Immuno^lo^ tumors. iii. PCR product indicated whether a given chromatin marker was present or absent. Predictive models based on the presence or absence of chromatin marker PCR products and built using methods such as XGBoost were used to narrow the list of candidate markers to 25. (**C**). Associations between blood chromatin markers and tumor JAV-Immuno levels. Univariate associations are shown between JAV-Immuno score and the status of 25 markers (present vs. absent) detected by PCR assay across 3 template dilution levels (1×, 2×, and 4×). The *x*-axis represents magnitude and direction of differences in mean JAV-Immuno levels. A Wilcoxon rank-sum test comparing JAV-Immuno scores between the presence and absence of each marker was performed at each dilution level. The *y*-axis represents the log_10_ of 2-sided *p*-values based on the test. Markers and dilutions that were selected in the final model are labeled. (**D**). Correlations between individual genes covered by selected CCM markers and JAV-Immuno signature scores in tumors are plotted as Pearson coefficients with log10 (*p* value). Genes with uncorrected *p* ≤ 0.001 are labeled.

**Figure 2 cancers-17-02332-f002:**
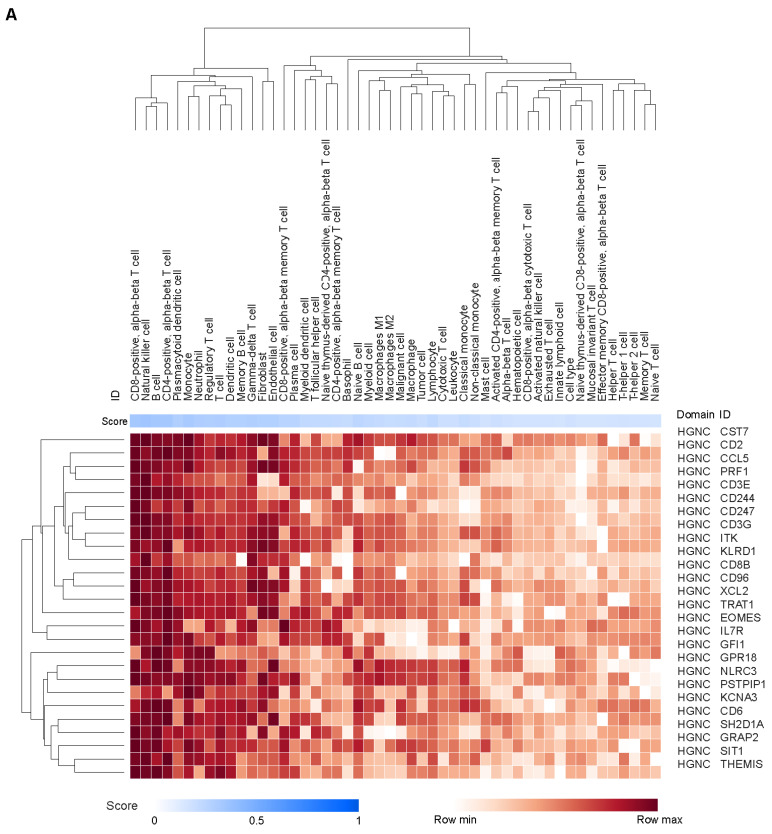
Cell types and functions of genes covered by the selected EpiSwitch CCM. The heat maps display aggregated associations between genes (rows) and cell types (columns in **A**,**B**) or functions (columns in **C**,**D**) across public data represented in the Data4Cure CURIE knowledge graph. The association score summarizes data and literature evidence from publicly available datasets and publications linking biomedical entities. Hierarchical clustering was applied to associations to generate the heat map. Association between cell types and genes in the JAV-Immuno signature (**A**) and the selected CCM set (**B**). Only genes with mean scores of ≥0.1 for at least 4 cell types are included. T cells and NK cells are strongly associated with JAV-Immuno, whereas genes in the selected CCMs display more diverse associations, including with endothelial cells, fibroblasts, B cells, and monocytes, as well as NK and T cells. Comparison of Gene Ontology biological processes associated with JAV-Immuno (**C**) and the selected CCM set (**D**). JAV-Immuno displays strong connections with T-cell processes, whereas selected CCM genes are associated with more diverse processes. CCM, chromatin conformation marker; NK, natural killer.

**Figure 3 cancers-17-02332-f003:**
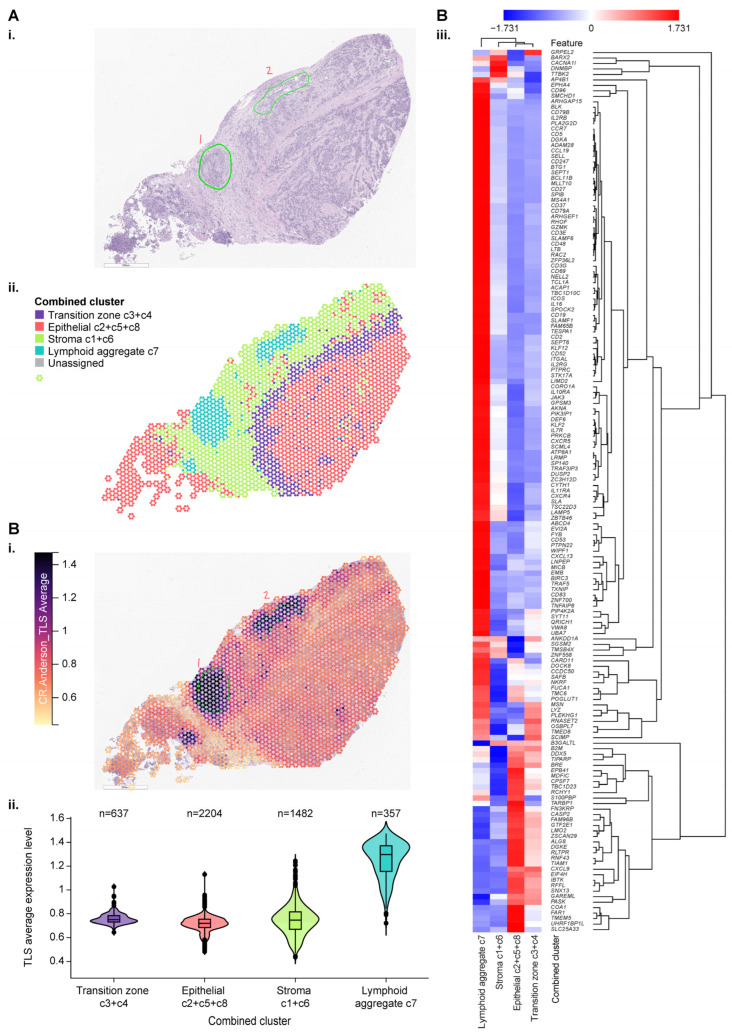
Spatial transcriptomic analysis in bladder cancer sections of marker set genes compared with JAV-Immuno genes. (**A**). (**i**). Hematoxylin and eosin stain of an index bladder cancer section (A1-1) showing the position of lymphoid aggregates annotated by a pathologist (labeled as 1 and 2). (**ii**). Spatial indexing transcriptomic data (Visium) from the same specimen processed by BayesSpace. Eight regions identified by BayesSpace were manually merged into 4 functional regions and annotated by majority cell type and histology. (**B**). (**i**). The A1-1 bladder specimen was annotated according to average expression of a published TLS signature [36] per spot. (**ii**). Average expression level of the TLS signature over the 4 main regions. (**iii**). Heat map depicting individual gene expression across the 4 regions. Color represents aggregation by means of values normalized by z-score. Note that most genes in the signature are expressed predominantly in the lymphoid aggregate region. (**C**). (**i**). A1-1 specimen annotated according to average expression of JAV-Immuno per spot. (**ii**). Average expression level of JAV-Immuno over the 4 main regions. (**iii**). Heat map depicting individual gene expression across the 4 regions. Note that most genes in the JAV-Immuno signature are also expressed predominantly in the lymphoid aggregate region. (**iv**). Concordance between JAV-Immuno genes localized to lymphoid aggregates in the A1-1 bladder specimen and TLS-high regions in the bladder/renal/breast cancer collection. (**D**). (**i**). Average expression level of the CCM model genes over the 4 regions. (**ii**). Heat map depicting individual gene expression across clusters based on distribution of expression between regions. Approximately half of the marker-set genes are expressed in lymphoid aggregates. (**iii**). The A1-1 bladder specimen annotated according to average expression of the CCM model genes per spot. Separate spatial images are shown for the epithelial and lymphoid regions. (**iv**). Concordance between genes localized to lymphoid aggregates in the A1-1 bladder specimen and TLS-high regions in the bladder/renal/breast cancer collection. CCM, chromatin conformation marker; TLS, tertiary lymphoid structures.

**Figure 4 cancers-17-02332-f004:**
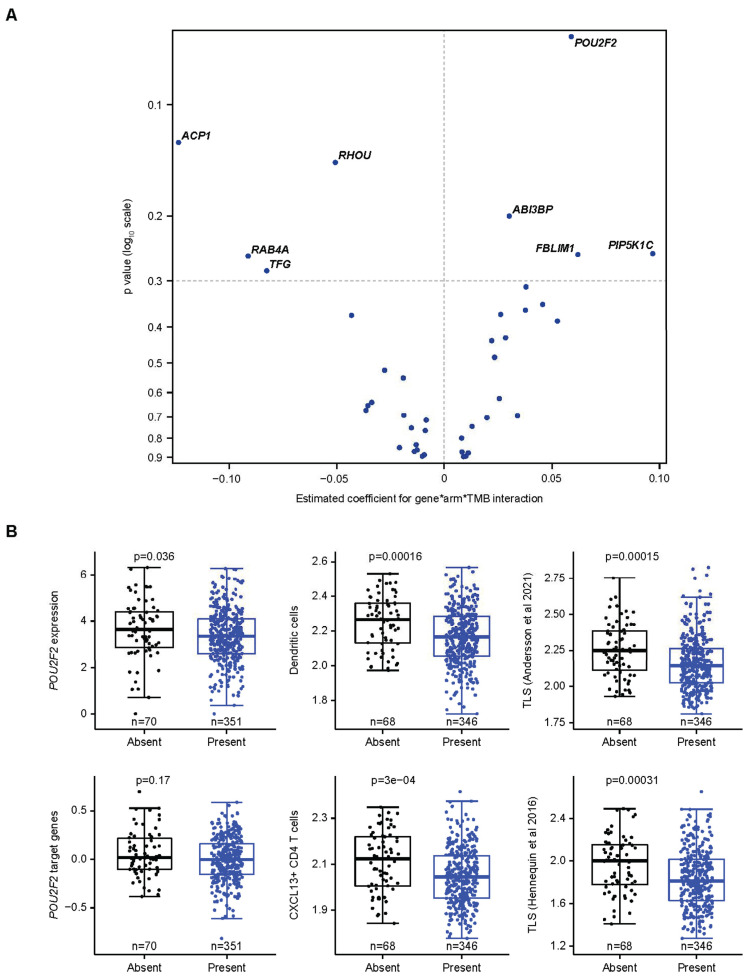
Modulation of interactions between TMB and OS by *POU2F2*. (**A**). Three-way interactions between *POU2F2* expression in tumor, TMB, and treatment on OS outcome. Continuous measures of TMB and gene expression (log_2_ TPM) were used in the analysis. Genes that were expressed in ≤50% of samples or had a coefficient of variation of ≤5 for expression were excluded from the analysis. The estimated coefficients for the 3-way interaction and the *p*-value on the *y*-axis were calculated based on a Cox model containing the gene*TMB*treatment interaction term and all the related lower-order terms. Note that *POU2F2* showed the strongest monogenic effect. (**B**). Presence of the blood marker encompassing *POU2F2* is associated with reduced expression of *POU2F2*, downstream target genes, and gene expression signatures characteristic of TLS. OS, overall survival; TLS, tertiary lymphoid structures; TMB, tumor mutation burden; TPM, transcript count per million [50,51].

**Table 1 cancers-17-02332-t001:** Compiled relationship between genes in the selected CCM set and JAV-Immuno, or entities in the CURIE knowledge graph. Values reported in the cell type, immune compartment, and immune process columns represent aggregated association scores between the individual genes and selected entities in the Data4Cure CURIE Knowledge Graph. The association score summarizes data and literature evidence from publicly available datasets and publications linking biomedical entities. CCM, chromatin conformation marker; MHC, major histocompatibility complex.

		Association Between Tumor Gene Expression and JAV-Immuno Score			Cell Type							Immune Compartment		Immune Process		
Gene	Gene Description	Statistic Value	*p*-Value	q-Value	Endothelial Cell	Fibroblast	B Cell	Monocyte	Natural Killer Cell	Dendritic Cell	Cytotoxic T Cell	Lymph Node	Germinal Center	MHC Class I Antigen Presentation	MHC Class II Antigen Presentation	Response to Stress
*POU2F2*	POU class 2 homeobox 2	0.8141921	2.75 × 10^−135^	4.96 × 10^−134^	0.09011	0.2449	0.2558	0.2563	0.228	0.2125	0.02845	0.09261	0.09931	0	0.04072	0.004806
*MBNL1*	Muscle blind-like splicing regulator 1	0.6073972	2.31 × 10^−58^	2.77 × 10^−57^	0.2185	0.222	0.2186	0.2325	0.07344	0.04999	0.02845	0	0	0.008677	0.008505	0.1479
*ABI3BP*	ABI family member 3 binding protein	0.5517624	2.09 × 10^−46^	1.88 × 10^−45^	0.212	0.2435	0.2236	0.1964	0.1889	0.1974	0.02832	0.07382	0	0	0	0
*VPS13C*	Vacuolar protein sorting 13 homolog C	0.4595461	6.43 × 10^−31^	4.63 × 10^−30^	0.07422	0.1806	0.05065	0.06594	0.06424	0.05733	0.02213	0	0	0	0	0
*SUCNR1*	Succinate receptor 1	0.4390713	4.48 × 10^−28^	2.69 × 10^−27^	0.2147	0.2118	0.1985	0.2108	0.2092	0.208	0.15	0.09239	0	0.003451	0.00712	0.0858
*RSAD2*	Radical S-adenosyl methionine domain containing 2	0.3267927	1.49 × 10^−15^	6.72 × 10^−15^	0.2266	0.2148	0.234	0.2137	0.2234	0.229	0.1652	0.09274	0.07864	0	0.06117	0.1613
*CMPK2*	Cytidine/uridine monophosphate kinase 2	0.2787244	1.47 × 10^−11^	5.89 × 10^−11^	0.2066	0.2194	0.2099	0.2243	0.1865	0.2063	0.1437	0.07898	0.06812	0	0.0499	0.08855
*PI4KA*	Phosphatidylinositol 4-kinase alpha	0.152742	0.000265	0.000954	0.2101	0.2087	0.2186	0.1967	0.197	0.04704	0.02591	0	0	0.01603	0	0
*RNF144A*	Ring finger protein 144A	0.1471635	0.000444	0.00145	0.09449	0.2008	0.1847	0.1624	0.06401	0.1495	0.08942	0.07085	0.01967	0	0	0
*CPEB1*	Cytoplasmic polyadenylation element binding protein 1	0.1458114	0.000502	0.0015	0.1894	0.2084	0.1893	0.1863	0.07279	0.05963	0	0	0	0	0	0.01358
*NPY4R*	Neuropeptide Y receptor Y4	0.1201768	0.00419	0.00944	0.04041	0.1401	0.1273	0.1208	0.1318	0.03498	0	0	0	0.003384	0	0
*ZNF573*	Zinc finger protein 573	0.1156841	0.00586	0.0113	0.06559	0.1173	0.1437	0.1506	0.1212	0.09401	0.04908	0.02246	0	0.005744	0	0
*SNAP29*	Synaptosome-associated protein 29	0.1154207	0.00598	0.0113	0.2049	0.1919	0.06076	0.06829	0.06407	0.04174	0.02845	0	0	0.006851	0.005367	0.1535
*ZNF781*	Zinc finger protein 781	0.1124934	0.00739	0.0121	0.1024	0.0865	0.1235	0.08621	0.09041	0.04321	0.005367	0.04858	0	0.003986	0.002624	0.000162
*SLC38A7*	Solute carrier family 38 member 7	0.0978701	0.0199	0.0286	0.1976	0.05722	0.192	0.06142	0.05257	0.03522	0.02277	0	0	0	0	0.08463
*DEDD2*	Death effector domain containing 2	0.0966533	0.0215	0.0297	0.08156	0.1777	0.1756	0.1643	0.1515	0.146	0.1052	0.06802	0	0	0.02047	0.06925
*C2CD4B*	C2 calcium-dependent domain containing 4B	0.0826085	0.0495	0.066	0.2162	0.1775	0.154	0.1431	0.1367	0.05737	0.01532	0.0178	0	0	0	0
*TMEM14E*	Transmembrane protein 14E, pseudogene	0.0678954	0.107	0.137	0.03418	0.0232	0.0338	0.03111	0.04544	0.02271	0.004312	0	0	0	0	0
*MMP16*	Matrix metallopeptidase 16	0.0639516	0.129	0.16	0.2134	0.2523	0.2056	0.2038	0.2085	0.06526	0.007903	0.09234	0	0	0	0
*LZTR1*	Leucine zipper-like transcription regulator 1	0.0550683	0.191	0.222	0.2021	0.1965	0.06215	0.1714	0.2034	0.03179	0.01318	0.08179	0	0.007257	0.008901	0.006873
*RPS17*	Ribosomal protein S17	0.0165515	0.694	0.757	0.2079	0.183	0.2076	0.2269	0.2022	0.179	0.1641	0	0	0.03428	0.07347	0.1058
*CNBD1*	Cyclic nucleotide binding domain containing 1	0.0040582	0.923	0.923	0.1355	0.1114	0.1151	0.1164	0.1073	0.09581	0.01741	0.04782	0	0.003208	0.003264	0.00432
*C2CD4A*	C2 calcium-dependent domain containing 4A	−0.004789	0.909	0.923	0.171	0.1587	0.1331	0.1213	0.1156	0.05684	0.004867	0.04788	0	0	0.003982	0.08443
*DCAF4L2*	DDB1 and CUL4-associated factor 4-like 2	−0.008794	0.835	0.884	0.06324	0.03736	0.06659	0.09815	0.1282	0.05735	0	0.06016	0	0.00528	0.01013	0.00442
*ZNF526*	Zinc finger protein 526	−0.048678	0.248	0.279	0.1291	0.1437	0.07571	0.06573	0.04032	0.03047	0.004803	0	0	0	0	0
*CNOT1*	CCR4-NOT transcription complex subunit 1	−0.055119	0.19	0.222	0.0678	0.21	0.2023	0.2026	0.03454	0.1721	0.01716	0	0	0.01848	0.003042	0.09112
*ZFP30*	ZFP30 zinc finger protein	−0.098922	0.0186	0.0279	0.1219	0.1227	0.1167	0.1308	0.1257	0.07336	0.02253	0	0	0	0	0
*ZNF607*	Zinc finger protein 607	−0.110554	0.00848	0.0133	0.04603	0.0471	0.06989	0.08244	0.1002	0.03778	0.0391	0.02276	0	0.004409	0.003307	0.01889
*GPRIN2*	G protein-regulated inducer of neurite outgrowth 2	−0.112585	0.00734	0.0121	0.06067	0.1465	0.1516	0.03462	0.1517	0.04801	0.00213	0.05896	0.02013	0	0.01973	0.00166
*TFG*	Trafficking from ER to golgi regulator	−0.114673	0.00631	0.0114	0.2094	0.1879	0.2205	0.2067	0.06115	0.02826	0.02845	0.08952	0	0.003158	0.00423	0.005551
*SERPIND1*	Serpin family D member 1	−0.116083	0.00569	0.0113	0.2145	0.1994	0.189	0.1905	0.05945	0.1672	0	0.09033	0	0	0.004157	0.1498
*CRKL*	CRK-like proto-oncogene, adaptor protein	−0.130329	0.00189	0.00454	0.2009	0.2036	0.2249	0.1969	0.2029	0.1667	0.0242	0.09401	0	0.02339	0.02136	0.02151
*AIFM3*	Apoptosis-inducing factor mitochondria-associated 3	−0.138936	0.000919	0.00236	0.19	0.1729	0.05	0.06931	0.0611	0.05562	0.006126	0.08887	0	0.0107	0.004403	0.1262
*ANXA8L1*	Annexin A8-like 1	−0.14415	0.000582	0.00161	0.1466	0.1922	0.1392	0.1212	0.1354	0.1276	0.02692	0.0607	0	0	0	0
*GOT2*	Glutamic-oxaloacetic transaminase 2	−0.381291	5.01 × 10^−21^	2.58 × 10^−20^	0.2346	0.2296	0.2188	0.2172	0.07141	0.1993	0.1728	0.09115	0.07472	0	0.006362	0.09641

**Table 2 cancers-17-02332-t002:** Associations between POU2F2 marker and OS.

Treatment	TMB	*POU2F2* Marker	No. of Patients (N = 457)	No. of Events	OS, Median (95% CI), Months	HR (*POU2F2* Marker Absent vs. Present) (95% CI)	*p*-Value
Avelumab plus BSC	≤Median	Absent	22	10	36.99 (18.17–NE)	0.46 (0.240–0.894)	0.0218
Avelumab plus BSC	≤Median	Present	112	82	17.77 (13.34–22.34)		
Avelumab plus BSC	>Median	Absent	11	7	19.25 (17.81–NE)	1.38 (0.623–3.048)	0.4281
Avelumab plus BSC	>Median	Present	98	48	35.12 (26.05–NE)		
BSC alone	≤Median	Absent	18	13	13.68 (8.8–NE)	1.14 (0.622–2.071)	0.6788
BSC alone	≤Median	Present	89	60	16.07 (10.25–24.18)		
BSC alone	>Median	Absent	21	14	14.78 (11.5–NE)	1.14 (0.635–2.044)	0.6608
BSC alone	>Median	Present	86	58	17.81 (13.54–26.64)		

BSC, best supportive care; HR, hazard ratio; NE, not estimable; OS, overall survival; TMB, tumor mutation burden.

## Data Availability

Any requests for data by qualified scientific and medical researchers for legitimate research purposes will be subject to Merck’s Data Sharing Policy. All requests should be submitted in writing to Merck’s data sharing portal (https://www.merckgroup.com/en/research/our-approach-to-research-and-development/healthcare/clinical-trials/commitment-responsible-data-sharing.html, accessed on 30 March 2025). When Merck has a co-research, co-development, co-marketing, or co-promotion agreement, or when the product has been out-licensed, the responsibility for disclosure might be dependent on the agreement between parties. Under these circumstances, Merck will endeavor to gain agreement to share data in response to requests.

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
