# Peer review of "Blood-Epigenetic Biomarker Associations with Tumor Immunophenotype in Patients with Urothelial Carcinoma from JAVELIN Bladder 100"

_cancers, 2025, doi:10.3390/cancers17142332_

Round 1
Reviewer 1 Report
Comments and Suggestions for Authors
The authors analyzed blood samples from 496 patients enrolled in the JAVELIN Bladder 100 study, in which avelumab demonstrated a benefit over best supportive care (BSC) when used as maintenance therapy following first-line platinum-based chemotherapy. They employed EpiSwitch, a validated 3D genomic biomarker platform, to identify chromatin conformation markers and to characterize the expression and function of genes associated with the highest-ranked markers. Several of these genes were involved in immune response pathways. POU2F2 was one of such genes.
Furthermore, the authors conducted spatial profiling using spatial transcriptomic analysis on bladder cancer tissue sections. Their findings suggest that chromatin conformation assays may serve as biomarkers for immune-oncology drugs in advanced bladder cancer.
This is an interesting and well-executed study. The manuscript is clearly presented, featuring four figures, two tables, six supplementary figures, two supplementary tables, and citing 50 references—an appropriate number for a study of this scale.
However, the discussion section appears relatively brief in comparison to the volume of data presented. It would benefit from a more comprehensive analysis and interpretation of the results.
Author Response
|
Reviewer 1 comments |
Response and proposed revisions |
1 |
The authors analyzed blood samples from 496 patients enrolled in the JAVELIN Bladder 100 study, in which avelumab demonstrated a benefit over best supportive care (BSC) when used as maintenance therapy following first-line platinum-based chemotherapy. They employed EpiSwitch, a validated 3D genomic biomarker platform, to identify chromatin conformation markers and to characterize the expression and function of genes associated with the highest-ranked markers. Several of these genes were involved in immune response pathways. POU2F2 was one of such genes. Furthermore, the authors conducted spatial profiling using spatial transcriptomic analysis on bladder cancer tissue sections. Their findings suggest that chromatin conformation assays may serve as biomarkers for immune-oncology drugs in advanced bladder cancer. This is an interesting and well-executed study. The manuscript is clearly presented, featuring four figures, two tables, six supplementary figures, two supplementary tables, and citing 50 references—an appropriate number for a study of this scale. However, the discussion section appears relatively brief in comparison to the volume of data presented. It would benefit from a more comprehensive analysis and interpretation of the results
|
We thank Reviewer 1 for the kind perspective. The discussion section has been updated to include a more detailed summary of the findings and possible avenues for further research. |

Reviewer 2 Report
Comments and Suggestions for Authors
- This study is an exploratory post hoc analysis based on the JAVELIN Bladder 100 trial. While chromatin conformation markers (CCMs) present potential, their clinical relevance remains unvalidated. Therefore, the findings are hypothesis-generating and not yet ready for clinical implementation.
- Although internal validation using cross-validation was performed, the absence of an independent external validation cohort raises concerns regarding the model’s generalizability and robustness.
- The initial selection of only 40 patients (20 high and 20 low JAV-Immuno) for training may introduce feature selection bias, which could affect the stability and reproducibility of the final model.
- The Cox regression models did not clearly adjust for potential clinical confounders such as age, sex, ECOG performance status, or tumor staging. These factors may affect survival outcomes and interfere with the interpretation of CCM associations.
- Several mapped genes, such as ZNF family members or pseudogenes, have poorly understood biological functions. Their roles in immune regulation remain unclear, weakening the mechanistic interpretation of the findings.
- The spatial transcriptomics analysis was based on a single specimen, without statistical comparisons or validation. The conclusions drawn regarding tertiary lymphoid structures (TLS) may be overinterpreted.
- While the study presents compelling exploratory data, some statements regarding clinical applicability—especially those involving POU2F2 as a predictive biomarker—appear overly optimistic without prospective trial confirmation.
- Table 1 comparisons are made between the full analysis set and the OBD (biomarker) subset, no statistical tests (e.g., chi-square or t-test) are provided to confirm whether differences are significant. This limits confidence in the statement that the biomarker subset is representative.
- Table 2 includes multiple genetic loci, marker IDs, gene names, and p-values across several interaction types (main, 2-way, 3-way), but lacks a clear legend or explanatory note on how these should be interpreted by non-specialist readers. Moreover, P-values are shown without adjusted values (e.g., FDR), which raises concerns about false discovery due to the large number of markers screened.
The English could be improved to more clearly express the research.
Author Response
|
Reviewer 2 comments |
Response and proposed revisions |
|
This study is an exploratory post hoc analysis based on the JAVELIN Bladder 100 trial. While chromatin conformation markers (CCMs) present potential, their clinical relevance remains unvalidated. Therefore, the findings are hypothesis-generating and not yet ready for clinical implementation.
|
We agree with the reviewer that findings from this exploratory post hoc analysis require further validation in independent datasets and demonstration of clinical utility. The first paragraph of the discussion section has been updated to provide a concise summary of the limitations in order to focus the readers’ attention on the emerging hypotheses and next steps. |
|
Although internal validation using cross-validation was performed, the absence of an independent external validation cohort raises concerns regarding the model’s generalizability and robustness.
|
The first paragraph in the discussion has been updated to discuss the validation in independent datasets as a necessary next step. |
|
The initial selection of only 40 patients (20 high and 20 low JAV-Immuno) for training may introduce feature selection bias, which could affect the stability and reproducibility of the final model.
|
The first paragraph in the discussion has been updated to discuss the confirmation of generalizability and robustness given potential biases in the training set as a necessary next step. |
|
The Cox regression models did not clearly adjust for potential clinical confounders such as age, sex, ECOG performance status, or tumor staging. These factors may affect survival outcomes and interfere with the interpretation of CCM associations.
|
The first paragraph in the discussion has been updated to discuss the further investigation of potential clinical confounders as a necessary next step. |
|
Several mapped genes, such as ZNF family members or pseudogenes, have poorly understood biological functions. Their roles in immune regulation remain unclear, weakening the mechanistic interpretation of the findings.
|
The third paragraph in the discussion has been updated to describe additional research investigating the links between the genes and the tumor phenotype. |
|
The spatial transcriptomics analysis was based on a single specimen, without statistical comparisons or validation. The conclusions drawn regarding tertiary lymphoid structures (TLS) may be overinterpreted.
|
The results section 3.4 describes assessment of JAV-Immuno and CCM genes in spatial transcriptomic data from a collection of specimens representing bladder, renal and breast cancer. The text has been modified to clarify the data type. Also, the second paragraph in the discussion has been rewritten to address the need for additional spatial profiling to confirm the generalizability of the findings. |
|
While the study presents compelling exploratory data, some statements regarding clinical applicability—especially those involving POU2F2 as a predictive biomarker—appear overly optimistic without prospective trial confirmation.
|
The first paragraph in the discussion has been updated to discuss the prospective demonstration of utility in clinical studies as a necessary next step. |
|
Table 1 comparisons are made between the full analysis set and the OBD (biomarker) subset, no statistical tests (e.g., chi-square or t-test) are provided to confirm whether differences are significant. This limits confidence in the statement that the biomarker subset is representative.
|
The reviewer appears to be referring to Supplementary Table 1, not main Table 1. The main text has been adjusted to remove the implication that statistical testing was performed. |
|
Table 2 includes multiple genetic loci, marker IDs, gene names, and p-values across several interaction types (main, 2-way, 3-way), but lacks a clear legend or explanatory note on how these should be interpreted by non-specialist readers. Moreover, P-values are shown without adjusted values (e.g., FDR), which raises concerns about false discovery due to the large number of markers screened.
|
The reviewer appears to be referring to Supplementary Table 2, not main Table 2. The legend contains a concise description and refers the readers to the Methods section for further details. The statistical limitations of the analyses are encompassed by the caveats discussed in the first paragraph of the discussion. |

Reviewer 3 Report
Comments and Suggestions for Authors
This manuscript must be revised following the specific comments/suggestions below before re-review. Specific comments are below.
- Try to use informative phrase or sentence for result section in the subtitle as taking home message. For example, “3.1. Study cohort and methodology” tell no information.
- Figure and supplemental figure fonts used are too small to see. When possible (space available), always use large fonts. Please to be creative. For example, in the current Figure 2 format, the figure 2ABCD can be much larger within the space left. And then the font can be much larger. Even there is no figure larger, font for the color label scale at the bottom of each of the subfigure section can be much larger, etc.
- Suddenly, “Table 15. CCMs in the final model ……”, All figures and tables should be in sequential order.
- Please keep the format consistent. For example, the format below is not clear to this review.
“CCM, chromatin conformation marker; NK, natural killer.
3.4. Spatial profiling
Spatial transcriptomic……”
- Additionally, this reviewer encourages these authors trying to show the data in a readers’ friendly format, when possible. For example, given the current Figure 2ABCD data format, whether it is possible for each section to insert a simplified gene/expression-fucntional association example in other format such as X/Y axis/histogram but not always use complicated heatmap format.
Generally speaking, fine.
Author Response
|
Reviewer 3 comments |
|
1. |
Try to use informative phrase or sentence for result section in the subtitle as taking home message. For example, “3.1. Study cohort and methodology” tell no information.
|
More descriptive subtitles have been applied throughout the manuscript. |
|
Figure and supplemental figure fonts used are too small to see. When possible (space available), always use large fonts. Please to be creative. For example, in the current Figure 2 format, the figure 2ABCD can be much larger within the space left. And then the font can be much larger. Even there is no figure larger, font for the color label scale at the bottom of each of the subfigure section can be much larger, etc.
|
The figures have been revised to improve readability. |
|
Suddenly, “Table 15. CCMs in the final model ……”, All figures and tables should be in sequential order. |
This typo has been removed. |
|
Please keep the format consistent. For example, the format below is not clear to this review.
“CCM, chromatin conformation marker; NK, natural killer. 3.4. Spatial profiling Spatial transcriptomic……”
|
We have corrected his formatting edit in the manuscript. |
|
Additionally, this reviewer encourages these authors trying to show the data in a readers’ friendly format, when possible. For example, given the current Figure 2ABCD data format, whether it is possible for each section to insert a simplified gene/expression-fucntional association example in other format such as X/Y axis/histogram but not always use complicated heatmap format.
|
We appreciate the reviewer’s recommendation. The heat map format has been retained since we consider the value of maintaining consistency across figures to justify the complexity. |

Reviewer 4 Report
Comments and Suggestions for Authors
The authors presented the work titled as "Blood-epigenetic biomarker associations with tumor immunophenotype in patients with urothelial carcinoma from JAVELIN Bladder 100". The overall work design and presentation is standard. There are few concerns which the authors need to address:
- The author should remove the orcid id from the main page.
- I will recommend to expand the details of JAVELIN Bladder 100 in introduction as a separate paragraph. I could see some highly relevant references specially from NEJM journal which will guide the authors in this regard.
- The first sentence of last paragraph of introduction section is confusing and needs to be rewritten.
- In section 2.2, the last sentence looks like something is missing and I am confused with "normalized to 5ng/mL".
- Will it be possible to include a pseudocode or workflow in method section for all the steps used to perform this study?
- on page 16, there is some strange text line: "
CCM, chromatin conformation marker; NK, natural killer.
" - For figure 3, the author could also use some proper clustering approach for the segmented image for different groups while I still consider it as impressive approach. I am attaching sample figure which is my own figure if author could do something similar to it (Not Mandatory : it is upto the authors only). Please see the attached file for help.

Author Response
Reviewer 4 comments |
|
The author should remove the orcid id from the main page.
|
The Orcid number was removed. |
I will recommend to expand the details of JAVELIN Bladder 100 in introduction as a separate paragraph. I could see some highly relevant references specially from NEJM journal which will guide the authors in this regard |
The description of the JAVELIN Bladder biomarker findings has been updated to provide a more comprehensive summary of the hypotheses emerging from that analysis that set the stage for the current work. |
The first sentence of last paragraph of introduction section is confusing and needs to be rewritten.
|
As suggested by the reviewer, this sentence has been updated. |
In section 2.2, the last sentence looks like something is missing and I am confused with "normalized to 5ng/mL".
|
The formatting error has been corrected. The phrase “of input DNA” has been added for clarification. |
Will it be possible to include a pseudocode or workflow in method section for all the steps used to perform this study?
|
Figure 1B includes a workflow to guide the readers through the processes used. |
on page 16, there is some strange text line: "CCM, chromatin conformation marker; NK, natural killer.”
|
The typo has been corrected, and the section header has been revised. |
For figure 3, the author could also use some proper clustering approach for the segmented image for different groups while I still consider it as impressive approach. I am attaching sample figure which is my own figure if author could do something similar to it (Not Mandatory: it is up to the authors only). Please see the attached file for help.
|
We appreciate the Reviewers’ suggestion. As noted in our response to Reviewer 3, the figure format was chosen to maintain consistency across displays. |

Round 2
Reviewer 2 Report
Comments and Suggestions for Authors
The author has revised the manuscript according to the suggestions.
Comments on the Quality of English LanguageThe English could be improved to more clearly express the research.
Reviewer 3 Report
Comments and Suggestions for Authors
None.